# LANGUAGE MODELING USING TENSOR TRAINS

## ABSTRACT

Tensor networks have previously been shown to have potential in language modeling in theory but lack practical evidence support. We propose a novel Tensor Train Language Model (TTLM) based on Tensor-Train decomposition. To show the usefulness of TTLM, we perform a principled experimental evaluation on real-world language modeling tasks, showing that our proposed variants, TTLM-Large and TTLM-Tiny, can be more effective than Vanilla RNNs with low-scale of hidden sizes. Also, we demonstrate the relationship between TTLM and Second-order Recurrent Neural Networks (RNNs), Recurrent Arithmetic Circuits, and Multiplicative Integration RNNs in the sense that the architectures of all of these are, essentially, special cases of that of TTLM.[1]

## 1 INTRODUCTION

A *language model* assigns probabilities of sequences of words from the vocabulary $V$; the number of texts increases exponentially w.r.t to length $N$. Hence the domain of a language model is, by definition, the exponential space $\mathbb{V}^N$. However, due to the vanilla exponential space being intractable, existing work tends to use recurrent or auto-regressive architectures to generate conditional probabilities based on the context (typically encapsulated as a fixed-length dense vector). This indeed simplifies the calculation.

Recently, researchers (Pestun & Vlassopoulos, 2017; Miller et al., 2021; Zhang et al., 2019) have reconsidered the view of language models as joint probabilities of text, as it leads to exponential representations in tensor space. Word connections could be preserved in the exponential tensor space when measuring joint probabilities. To deal with the exponential space complexity, a mathematical tool called 'tensor network' [2] has been used to reduce the exponential space of language modeling to a tractable one (Pestun & Vlassopoulos, 2017). However, the so-called 'tensor network language model' in Pestun & Vlassopoulos (2017) is currently a concept that needs to be proved practically.

As proof-of-concept work, we derive a Tensor Train Language Model (TTLM) (the simplest tensor network). Technically, we represent a sentence based on the exponential semantic space constructed by the tensor product of word representations. The probability of the sentence is obtained by the inner product of two high-dimensional tensors: the input $\Phi(X)$ and the global coefficients $\mathcal{A}$.

Under the framework of TTLM, we propose two variants: TTLM-Tiny and TTLM-Large. Also, we clarify the relationship between the proposed TTLM and a series of Recurrent Neural Networks (RNNs) (i.e., Second-order RNNs (Goudreau et al., 1994), Recurrent Arithmetic Circuits (RACs) (Levine et al., 2018), and Multiplicative Integration RNNs (MI-RNNs) (Wu et al., 2016)). These connections open a new eye to understanding RNNs and give some natural implementations for TTLM.

We benchmark these TTLM variants and analyze the difference in their working mechanism and behaviors. Experimental results on language modeling tasks show that our TTLM variants could outperform than Vanilla-RNNs under the same training setting. These demonstrate the feasibility of TTLM.

The main contributions of our work can be summarized as follows:

---

[1]The code is available at `https://github.com/tensortrainlm/tensortrainlm`

[2]Tensor networks are, roughly, decompositions of large tensors into sets of smaller tensors and have been employed in physics, mathematics, and machine learning (Sun et al., 2020; Novikov et al., 2015; Cohen et al., 2016; Stoudenmire & Schwab, 2016b; Cheng et al., 2019; Novikov et al., 2016; Selvan & Dam, 2020).

1. We propose a novel Tensor Train Language Model, as a first attempt to apply tensor networks on real-world language modeling tasks.

2. We propose two novel TTLM variants, TTLM-Large and TTLM-Tiny, and theoretically demonstrate the relationship between TTLM and a series of existing RNNs.

3. Compared to Vanilla-RNNs on WikiText-2 and PTB datasets, TTLM-Large reduces perplexity by 14.3 and 16.0, respectively, and TTLM-Tiny reduces perplexity by 1.7 and 8.5, respectively.

## 2 RELATED WORK

Previous studies on tensor networks in machine learning have mainly been devoted to analyzing the theoretical properties of neural networks. A better understanding of feed-forward, convolutional and recurrent architectures has been gained, including compression parameters (Novikov et al., 2015), expressive power (Cohen et al., 2016; Cohen & Shashua, 2016; Khrulkov et al., 2018), and depth efficiency for long-term memory (Levine et al., 2018).

Focusing on natural language modeling, certain studies have tensorized existing network architectures (Novikov et al., 2015), while few studies have applied tensor networks alone as a language model. To the best of our knowledge, tensor network language models have remained a theoretical proposal instead of an empirical model (Pestun & Vlassopoulos, 2017; Pestun et al., 2017). Perhaps, Miller et al. (2021) is the only other work that uses tensor networks for probabilistic sequence modeling, while it fails to scale up its model for real-world sequence modeling tasks. We first derive a tensor network language model in the way that it can be applied to real-world language modeling datasets, while its variants outperform Vanilla RNNs with lower-scale hidden sizes.

## 3 PRELIMINARIES

We briefly recapitulate basic notions and notations [3]; full technical introductions can be found in standard textbooks (e.g., Bi et al. (2022); Itskov (2009)).

**Notation.** For the purposes of this paper, every *tensor* **A** is a multidimensional array of elements (called *components*) of $\mathbb{R}$, each denoted by its integer coordinates in the array; e.g., for a two-dimensional array, the component at position $i, j \in \mathbb{N}$ is denoted $A_{ij}$. The *order* of a tensor is how many indices it has (e.g., a vector $v$ is a first-order tensor, a matrix $M$ is a second-order tensor, etc.). The *dimension* of a tensor refers to the number of values that a particular index (or so-called *mode*) can take, e.g., the dimension of $\mathbf{B} \in \mathbb{R}^{I_1 \times I_2 \times I_3}$ is $I_1 \times I_2 \times I_3$.

**Tensor product.** For two tensors $\mathbf{C} \in \mathbb{R}^{I_1 \times \cdots \times I_j}$ (order $j$) and $\mathbf{D} \in \mathbb{R}^{I_{j+1} \times \cdots \times I_{j+k}}$ (order $k$), their *tensor product* is denoted by $\otimes$ and return a tensor $E_{i_1 \cdots i_{j+k}} = C_{i_1 \ldots i_j} \cdot D_{i_{j+1} \cdots i_{j+k}}$ (order $j + k$).

**Generalized inner product.** For two tensor $\mathbf{X}, \mathbf{Y} \in \mathbb{R}^{I_1 \times I_2 \times \cdots \times I_N}$ of the same size, their *inner product* is defined as $\langle \mathbf{X}, \mathbf{Y} \rangle = \sum_{i_1=1}^{I_1} \sum_{i_2=1}^{I_2} \cdots \sum_{i_N=1}^{I_N} X_{i_1,i_2,\ldots,i_N} Y_{i_1,i_2,\ldots,i_N}$. For two tensors $\mathbf{X} \in \mathbb{R}^{I_1 \times I_2 \times \cdots \times I_N \times I_x}$ and $\mathbf{Y} \in \mathbb{R}^{I_1 \times I_2 \times \cdots I_N \times I_y}$ sharing $N$ modes of the same size, the "generalized inner product" defined in (Kossaifi et al., 2020) is calculated as

$$\langle \mathbf{X}, \mathbf{Y} \rangle_N = \sum_{i_1=1}^{I_1} \sum_{i_2=1}^{I_2} \cdots \sum_{i_N=1}^{I_N} X_{i_1,i_2,\ldots,i_N} Y_{i_1,i_2,\ldots,i_N}$$

with $\langle \mathbf{X}, \mathbf{Y} \rangle_N \in \mathbb{R}^{I_x \times I_y}$.

## 4 LANGUAGE MODELING USING TENSOR TRAINS

We introduce a language model in tensor space in Sec. 4.1. We define our *general Tensor Train Language Model* in Sec. 4.2, and its special case, TTLM, in Sec. 4.3.

---

[3] Most of the notations here follow the textbook *Deep Learning* Goodfellow et al. (2016).

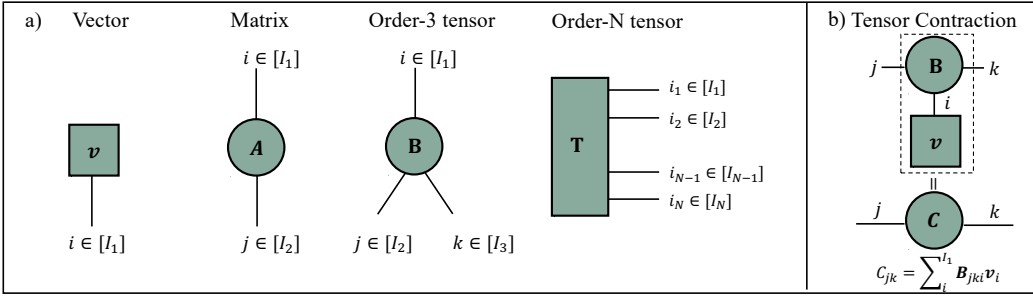

Figure 1: A quick introduction to *tensor diagram notation*. There are two rules of tensor diagrams. (1) tensors are notated by solid shapes with a number of 'legs' corresponding to their indices. (2) connecting two index lines implies a *contraction* or summation over the connected indices. In this paper, we augment our equations with these diagrams to make them easier to visualize.

## 4.1 Language models in a tensor space

Natural language typically has complex dependencies between features (e.g., tokens or words) (Hou et al., 2013)[4] that are not captured well by standard methods such as feature concatenation. One could also see a similar interaction between any arbitrary features in factorization machines (Rendle, 2010). Suppose a given text consists of $N$ words $X = [x^{(1)}, x^{(2)}, \cdots, x^{(N)}]$ and let $\boldsymbol{f}_i \in \mathbb{R}^{I_i}$ be a feature extractor (it can be one-hot encoding or word embedding). We now define a representation of $X$ designed to capture these dependencies:

$$\Phi(X) = \boldsymbol{f}_1(x^{(1)}) \otimes \boldsymbol{f}_2(x^{(2)}) \cdots \otimes \boldsymbol{f}_N(x^{(N)})$$
$$= \bigotimes_{i=1}^{N} \boldsymbol{f}_i(x^{(i)}) \tag{1}$$

where the tensor space is $\mathbb{R}^{I_1} \otimes \mathbb{R}^{I_2} \otimes \cdots \otimes \mathbb{R}^{I_N}$. Each component of $\boldsymbol{f}_i$ represents independent meaning-bearing units, such as morphemes or latent factors. For simplicity, we assume that a text shares the same one-hot encoding $\boldsymbol{f}(x^{(t)}) \in \mathbb{R}^{|V|}$ in later sections. Consequently, $\Phi(X)$ is a $|V|^N$-dimensional tensor that records all possible combinations of words in $X$.

Inspired by Kossaifi et al. (2020); Zhang et al. (2019), we define a tensor regression model to obtain the estimated probability for each text $X$:

$$p(X) = \langle \mathcal{A}, \Phi(X) \rangle$$
$$= \sum_{i_1, i_2, \cdots, i_N=1}^{|V|} \mathcal{A}_{i_1, \cdots, i_N} \cdot \Phi(X)_{i_1, \cdots, i_N} \tag{2}$$

where $\langle \cdot \rangle$ denotes the inner product of two same-sized tensors, and $\mathcal{A}$ is a regression weight tensor of the same shape as $\Phi(X)$ in the tensor space $\mathbb{V}^{\otimes N} = \underbrace{\mathbb{V} \otimes \cdots \otimes \mathbb{V}}_{N}$ where $\mathbb{V}$ refers to $\mathbb{R}^{|V|}$. Similar functions were considered in Novikov et al. (2016); Stoudenmire & Schwab (2016a); Khrulkov et al. (2018); Zhang et al. (2019).

## 4.2 General Tensor Train Language model

Suppose the sequence of indices of words in the text $X$ is $w_1, w_2, \cdots, w_N$, where $w_i \in \{1, 2, \cdots, |V|\}$ and its corresponding weight in $\mathcal{A}$ is denoted as $\mathcal{A}_{w_1 w_2 \cdots w_N}$. We use TT decomposition to represent $\mathcal{A}_{w_1 w_2 \cdots w_N}$ in the TT format (Oseledets, 2011) as follows:

---

[4]Such dependencies (including collocation) have been viewed as an analogy of entanglement (Hou et al., 2013).

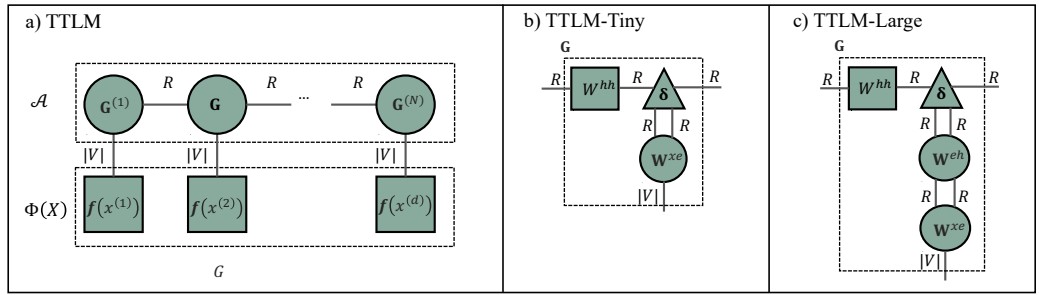

Figure 2: a) Tensor Train Language Model based on Eq. 5. b) TT core of TTLM-Tiny. c) TT core of TTLM-Large. The dashed line in the square represents $\mathcal{A}$, $\Phi(X)$, or $\mathbf{G}$. Note that the only difference between TTLM-Large and TTLM-Tiny is whether to use tensor $\mathbf{W}^{eh}$.

$$
\begin{aligned}
\mathcal{A}_{w_1 w_2 \ldots w_N} &= \underbrace{\mathbf{G}^{(1)}_{:,w_1}}_{1 \times R_1} \underbrace{\mathbf{G}^{(2)}_{:,w_2,:}}_{R_1 \times R_2} \cdots \underbrace{\mathbf{G}^{(N)}_{:,w_N}}_{R_{N-1} \times 1} \\
&= \sum_{\alpha_1=1}^{R_1} \sum_{\alpha_2=2}^{R_2} \cdots \sum_{\alpha_{N-1}=1}^{R_{N-1}} G^{(1)}_{w_1 \alpha_1} G^{(2)}_{\alpha_1 w_2 \alpha_2} \cdots G^{(N)}_{\alpha_{N-1} w_N}
\end{aligned}
\tag{3}
$$

where the tensors $\mathbf{G}^{(t)} \in \mathbb{R}^{R_{t-1} \times |V| \times R_t}$ ($t = 1, ..., d$, $R_0 = R_N = 1$ by definition) are called *TT cores*.

We now combine $\Phi(X)$ and $\mathcal{A}$ in the TT format to define *general TTLM*. The elements of $\mathbf{G}^{(t)}_{:,w_t,:}$ in Eq. 3 can be represented as:

$$
G^{(t)}_{\alpha_{t-1} w_t \alpha_t} = \sum_{i=1}^{|V|} f(x^{(t)})_i G^{(t)}_{\alpha_{t-1} i \alpha_t}
\tag{4}
$$

where each $\boldsymbol{f}(x^{(t)})$ is a one-hot vector having $w_t = 1$ for at most one $t$, and has zeros elsewhere. Therefore, one-hot encoding enables us to integrate the input data into the TT format of $\mathcal{A}$ by inserting Eq. 4 into Eq. 3:

$$
\begin{aligned}
\mathcal{A}_{w_1 w_2 \ldots w_N} &= \sum_{i_1, \cdots, i_N=1}^{|V|} \sum_{\alpha_1=1}^{R_1} \cdots \sum_{\alpha_{N-1}=1}^{R_{N-1}} f(x^{(1)})_{i_1} G^{(1)}_{i_1 \alpha_1} f(x^{(2)})_{i_2} G^{(2)}_{\alpha_1 i_2 \alpha_2} \cdots f(x^{(N)})_{i_N} G^{(N)}_{\alpha_{N-1} i_N} \\
&= \sum_{\alpha_1=1}^{R_1} \sum_{\alpha_2=2}^{R_2} \cdots \sum_{\alpha_{N-1}=1}^{R_{N-1}} G^{(1)}_{w_1 \alpha_1} G^{(2)}_{\alpha_1 w_2 \alpha_2} \cdots G^{(N)}_{\alpha_{N-1} w_N} \\
&= \langle \mathcal{A}, \Phi(X) \rangle = p(X)
\end{aligned}
\tag{5}
$$

where $\Phi(X) = \bigotimes_{i=1}^{N} \boldsymbol{f}(x^{(i)})$. The difference between Eq. 5 and Eq. 3 is that Eq. 5 has combined $\mathcal{A}$ and $\Phi(X)$ in the low-dimensional form. This is because that Eq. 3 can compute the elements of $\mathcal{A}$ (Oseledets, 2011), and because $\Phi(X)$ here is the tensor product of one-hot vectors, so that Eq. 5 can compute Eq. 2. Further, since Eq. 5 now has input data (one-hot vectors) and weights (TT cores), we name Eq. 5 as our *general TTLM*.

## 4.3 TTLM

Here we consider a special class of *general TTLM*. Despite its site-dependent TT cores $\mathbf{G}^{(t)}$ potentially giving it more expressiveness for language modeling, this property currently generates unnecessary obstacles to its applicability, like the choice of $R_t$. We here provide a detailed explanation of its special case: TTLM.

**Definition.** Suppose all the intermediate TT cores are equal to each other $\mathbf{G} = \mathbf{G}^{(2)}, \ldots, \mathbf{G}^{(N-1)} \in \mathbb{R}^{R \times |V| \times R}$ and $\mathbf{G}^{(1)} = \mathbf{G}^{(N)} \in \mathbb{R}^{|V| \times R}$ in Eq. 5. Then, TTLM is defined as follows:

$$
p(X) = \sum_{i_1, \cdots, i_N = 1}^{|V|} \sum_{\alpha_1, \cdots, \alpha_{N-1} = 1}^{R} f(x^{(1)})_{i_1} G^{(1)}_{i_1 \alpha_1} f(x^{(2)})_{i_2} G_{\alpha_1 i_2 \alpha_2} \cdots f(x^{(N)})_{i_N} G^{(N)}_{\alpha_{N-1} i_N}
$$

$$
= \sum_{\alpha_1, \cdots, \alpha_{N-1} = 1}^{R} G^{(1)}_{w_1 \alpha_1} G_{\alpha_1 w_2 \alpha_2} \cdots G_{\alpha_{N-2} w_{N-1} \alpha_{N-1}} G^{(N)}_{\alpha_{N-1} w_N} \tag{6}
$$

where its tensor diagram notation is shown in Figure 2a.

**Recursive information.** We recursively unfold the calculation of TTLM in Eq. 6 and find that $\mathbf{G}$ has two sources of "input": the information from the previous recursive unfolding, and the input data $\boldsymbol{f}(x^{(t)})$ (see Eq. 15 for a detailed version). From this perspective, $\mathbf{G}$ acts as a bilinear map $\mathbf{G} : \mathbb{R}^{|V|} \times \mathbb{R}^R \to \mathbb{R}^R$, and we can regard the information in the previous step as a hidden state $h^{(t)}_{\text{TTLM}}$, given by:

$$
h^{(t)}_{\text{TTLM}} = \boldsymbol{f}(x^{(t)})^T \mathbf{G} h^{(t-1)}_{\text{TTLM}} \tag{7}
$$

where $\boldsymbol{f}(x^{(t)})$, $\mathbf{G}$, and $h^{(t-1)}_{\text{TTLM}}$ are contracted together (we permute the indices of $\mathbf{G}$ from $\mathbb{R}^{R \times |V| \times R}$ to $\mathbb{R}^{|V| \times R \times R}$ which does not change the number of indices).

**Recursive Probability Computation.** Here, we here provide further details about the process of computing $p(X)$ by TTLM in practice.

In language modeling, $p(X)$ is often decomposed using the chain rule (Bahl et al., 1983) as follows:

$$
p(X) = \prod_{t=1}^{N} p(x^{(t)} | x^{(1:t-1)})
$$

where $x^{(1:t-1)}$ denotes the text $[x^{(1)}, x^{(2)}, \cdots, x^{(t-1)}]$. At time $t$, the output prediction of a model, $\boldsymbol{y}^{(t)} \in \mathbb{V}$, is a probability distribution of word $x^{(t)}$ given $x^{(1:t-1)}$.

In TTLM, we define $\boldsymbol{y}^{(t)}$ as follows:

$$
\boldsymbol{y}^{(t)} = \text{softmax}\left( \mathbf{G}^{(t)} h^{(t-1)}_{\text{TTLM}} \right) \tag{8}
$$

where $\mathbf{G}^{(t)} \in \mathbb{R}^{|V| \times R}$ is the last TT core in TT format

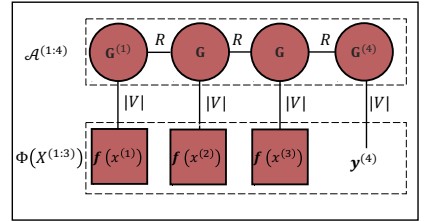

Figure 3: Recursive calculation of conditional probability in TTLM. Here we provide an example that given the text $x^{(1:3)}$, $\boldsymbol{y}^{(4)} = \text{softmax}(\mathbf{G}^{(4)} h^{(3)}_{\text{TTLM}})$ where $\boldsymbol{y}^{(4)} \in \mathbb{V}$ is the probability distribution of word $x^{(4)}$.

at time $t$. Such a definition is the same as that of RNNs, which use hidden states and a weight matrix to calculate word probabilities. Fig. 3 provides a simple example.

We can derive the definition of $\boldsymbol{y}^{(t)}$ in high-dimensional space, if we substitute $h^{(t-1)}_{\text{TTLM}}$ in Eq. 8 by Eq. 6 and Eq. 7:

$$
\boldsymbol{y}^{(t)} = \text{softmax}\left( \sum_{i_1, \cdots, i_{t-1} = 1}^{|V|} \sum_{\alpha_1, \cdots, \alpha_{t-1} = 1}^{R} f(x^{(1)})_{i_1} G^{(1)}_{i_1 \alpha_1} f(x^{(2)})_{i_2} G_{\alpha_1 i_2 \alpha_2} \cdots \mathbf{G}^{(t)}_{\alpha_{t-1}} \right) \tag{9}
$$

$$
= \text{softmax}\left( \langle \mathcal{A}^{(1:t)}, \bigotimes_{i=1}^{t-1} \boldsymbol{f}(x^{(i)}) \rangle_{t-1} \right) \tag{10}
$$

$$
= \text{softmax}\left( \langle \mathcal{A}^{(1:t)}, \Phi(X^{(1:t-1)}) \rangle_{t-1} \right) \tag{11}
$$

where $\mathcal{A}^{(1:t)} \in \mathbb{V}^{\otimes t}$, $\Phi(X^{(1:t-1)}) \in \mathbb{V}^{\otimes t-1}$ and $\langle \cdot \rangle_{t-1}$ denotes the "generalized inner product defined" in Preliminary. Note that Eq. 9 is the low-dimensional form of Eq. 11, similarly to the relationship between Eq. 5 and Eq. 2.

By these definitions, there are some interesting properties of TTLM. 1) We can use *teacher forcing* (Jurafsky, 2000) to learn parameters of TT cores. 2) The hidden-to-output tensor $\mathbf{G}^{(t)}$ is defined to be the same as the input-to-hidden tensor $\mathbf{G}^{(1)}$. 3) $\mathbf{G}$ and $\mathbf{G}^{(t)}$ have no parameters in common. We provide a detailed explanation of the relationship between different TT cores in Appendix A.

## 5 TTLM VARIANTS

To show the versatility and practical applicability of the TTLM framework, we now propose two new variants: TTLM-Large and TTLM-Tiny in Sec. 5.1. In Sec. 5.2, we briefly summarize the relationship between TTLM and some widely-used RNNs.

### 5.1 NEW VARIANTS: TTLM-LARGE AND TTLM-TINY

The TT core $\mathbf{G}$ in TTLM is an entire third-order tensor. In the two variants, we decompose $\mathbf{G}$ into several separate tensors without violating the TT format, as shown in Fig. 2b and Fig. 2c. We define TTLM-Tiny and TTLM-Large as follows:

$$
\begin{aligned}
h_{\text{Tiny}}^{(t)} &= \boldsymbol{f}(x^{(t)})^T \mathbf{W}^{xe} \boldsymbol{\delta} \boldsymbol{W}^{hh} h_{\text{Tiny}}^{(t-1)} \\
h_{\text{Large}}^{(t)} &= \boldsymbol{f}(x^{(t)})^T \mathbf{W}^{xe} \mathbf{W}^{eh} \boldsymbol{\delta} \boldsymbol{W}^{hh} h_{\text{Large}}^{(t-1)}
\end{aligned}
\tag{12}
$$

where $\mathbf{W}^{xe} \in \mathbb{R}^{|V| \times R \times R}$ is the input-to-hidden tensor ; $\mathbf{W}^{eh} \in \mathbb{R}^{R \times R \times R \times R}$; and $\boldsymbol{\delta} \in \mathbb{R}^{R \times R \times R \times R}$ is a fourth-order diagonal tensor such that $\delta_{ijkl} = 1$ iff the $i = j = k = l$, and $\delta_{ijkl} = 0$ otherwise.

The relationship between our proposed models and TTLM is as follows: $\mathbf{W}^{xe}$ in both models take the same role as $\mathbf{G}^{(t)}$ in TTLM (i.e. input-to-hidden and hidden-to-output), while $\mathbf{G} = \mathbf{W}^{xe} \boldsymbol{\delta} \boldsymbol{W}^{hh}$ in TTLM-Tiny and $\mathbf{G} = \mathbf{W}^{xe} \mathbf{W}^{eh} \boldsymbol{\delta} \boldsymbol{W}^{hh}$ in TTLM-Large.

As in RNNs, we compute the conditional probability recursively for TTLM-Large and TTLM-Tiny as:

$$
\boldsymbol{y}^{(t)} = \text{softmax}(\mathbf{V}\mathbf{P}\boldsymbol{h}^{(t)})
\tag{13}
$$

where $\mathbf{V} \in \mathbb{R}^{R \times |V| \times R}$ is an output embedding tensor, and $\mathbf{P} \in \mathbb{R}^{R \times R \times R}$ is a projector tensor. Then we tie the input tensor $\mathbf{W}^{xe}$ to the output embedding tensor $\mathbf{V}$ (we provide a detailed explanation in Appendix C).

One obvious advantage of our models is to utilize information from the hidden layer and input data separately. Such interaction, particularly TTLM-Tiny, can potentially avoid overfitting, similarly to Wu et al. (2016) where *multiplication integration* between two sources of "input" can outperform many other methods. In Sec 6.2, we provide relevant experimental evidence.

### 5.2 EXISTING TTLM VARIANTS

Given the fact that TT scores of TTLM can vary, Appendix B provides a detailed illustration that three existing models, namely second-order RNNs, RACs and MI-RNNs can be considered as one of the "special" implementations of TTLM.

We briefly summarize the differences between the three models: 1) Second-order RNNs use the third-order $\mathbf{T}$ as the TT cores with an activation function given Eq. 14; 2) RACs use $\boldsymbol{W}^{hx} \odot \boldsymbol{W}^{hh}$ as the TT cores given Eq. 18; 3) MI-RNNs use $\boldsymbol{W}^{hx} \odot \boldsymbol{W}^{hh}$ as the TT cores with an activation function given Eq. 19.

Along with our two proposed models, we study the experimental performance of second-order RNNs, RACs and MI-RNNs compared to TTLM-Large and TTLM-Tiny in Section 6.

## 6 EXPERIMENTAL EVALUATION

To further understand the properties of TTLM variants, we now investigate the effectiveness of TTLM-Large and TTLM-Tiny compared to Second-order RNNs, RACs, MI-RNNs, and Vanilla-RNNs. We conduct experiments from two distinct perspectives: (1) The *rank* of TT decomposition

| Model | WikiText-2 | | PTB | | Hidden (Rank) | Layer | Embed Size |
|-------|-----------|-----|------|-----|------|-------|------------|
| | Params | PPL | Params | PPL | | | |
| RNN (Mikolov & Zweig, 2012) | - | - | - | 124.7 | 300 | 1 | - |
| LSTM (Zaremba et al., 2014) | - | - | - | 114.5 | 200 | 2 | - |
| LSTM (Grave et al., 2016) | - | 99.3 | - | 82.3 | 1024 | 1 | - |
| LSTM (Merity et al., 2017) | - | 100.9 | - | 80.6 | 650 | 2 | - |
| Vanilla-RNNs* | 11.6M | 96.6 | 4.0M | 115.3 | 20 | 1 | 400 |
| Second-order RNNs* | 11.8M | 96.0 | 4.2M | 108.2 | 20 | 1 | 400 |
| RACs* | 11.6M | 97.6 | 4.0M | 116.8 | 20 | 1 | 400 |
| MI-RNNs* | 11.6M | 99.6 | 4.0M | 119.1 | 20 | 1 | 400 |
| TTLM | 12.2M | 546.4 | 4.2M | 559.8 | 20 | 1 | 400 |
| TTLM-Tiny | 11.6M | 94.9 | 4.0M | 106.8 | 20 | 1 | 400 |
| TTLM-Large | 11.8M | **82.3** | 4.2M | **99.3** | 20 | 1 | 400 |

Table 1: PPL evaluation on test set on WikiText-2 and PTB. Models tagged with $*$ indicate that they are re-implemented by ourselves. The symbol "−" means these data are not available in their original paper. Params are the training parameters, the details are in Appendix C

has been proved to be the dimension of the hidden states of RNNs (Khrulkov et al., 2018). Here, we study the influence of rank on the effectiveness of our TTLM variants in Sec 6.2. (2) In Sec.6.3, we analyse the influence of non-linearity for TTLM variants.

## 6.1 EXPERIMENTAL SETTING

**Tasks, Datasets, and Metrics.** We conduct experiments on word-level language model datasets: English Penn Treebank (PTB), which consists of 929k training words, 73k validation words, and 82k test words. It has 10k words in its vocabulary (Marcinkiewicz, 1994). The WikiText-2 dataset (Merity et al., 2016) is derived from Wikipedia articles and consists of 2088k training words, 217k validation words, 45k test words, and a vocabulary of over 30,000 words. We compare these models on the language modelling task, evaluated by the Perplexity (PPL) (Meister & Cotterell, 2021): the lower the perplexity, the better the model.

**Baselines.** *Vanilla RNNs*, *Second-RNNs*, *RACs* and *MI-RNNs* are our baselines. We also provide some original results of RNN-based models as references (Mikolov & Zweig, 2012; Zaremba et al., 2014; Grave et al., 2016; Merity et al., 2017). The implementation details are introduced in Appendix C.

**Hyperparameters.** To compare the effectiveness of comparable models in the same scale: 1) We set the rank/hidden size of TTLM variants/Vanilla-RNNs as [5, 10, 20, 25, 30, 35, 40, 45, 50]. The embedding size of these models is the squared number of hidden sizes/ranks. 3) To avoid the impact of the large embedding size on the model performance, we also provide several common choices of embedding size in Vanilla-RNNs by setting its embedding size as [100, 200, 300] (we name them as RNNs-100, RNNs-200, RNNs-300 correspondingly and use them in Fig. 5). 4) The random seed is fixed to ensure the experimental results are not influenced by initializing the weights. 5) We train all models for 50 epochs and choose the best model in the validation set to predict the result in the test set.

## 6.2 RANK AND EFFECTIVENESS ANALYSIS

The rank of the TT format has been used to explain the expressive power or long-term memory capacity of RNNs (Khrulkov et al., 2018; Levine et al., 2018). However, the relationship between rank and effectiveness in language modelling has yet to be shown practically. Later, we will evaluate the effectiveness of our models w.r.t rank.

**Effectiveness.** **(1)** Fig. 4 shows the influence of the rank on our models based on the validation PPL. The validation PPL of TTLM-Large drops down at the early training step but easily increases when the rank increases. In contrast, the validation PPL of TTLM-Tiny stably decreases as the rank increases. **(2)** Compared to Vanilla-RNNs, the influence of the rank on our models based on the test

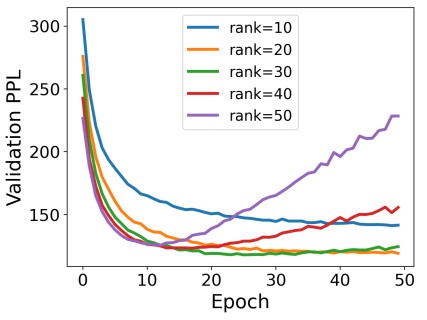 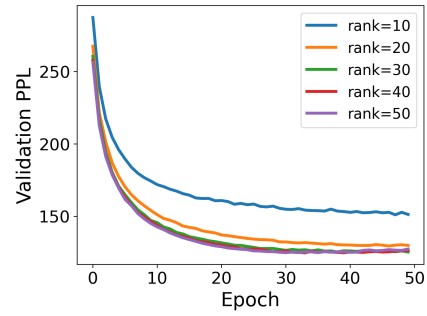

(a) Validation perplexity on TTLM-Large     (b) Validation perplexity on TTLM-Tiny

Figure 4: Rank analysis for the TTLM-Large and TTLM-Tiny on PTB.

PPL is shown in Fig. 5. Our models outperform Vanilla-RNNs on all used parameter settings. **(3)** Table 1 provides a supplementary example to show the comparison of our models with baselines and some references. TTLM-Large and TTLM-Tiny are more effective than baselines.

**Overfitting.** Based on Fig. 4a and Fig. 4b, we find that TTLM-Large is more prone to overfitting than TTLM-Tiny. As the only difference between the two models is $\mathbf{W}^{eh}$, this suggests that the simpler parameterization of the TT cores, the more easily the model avoids overfitting. This finding is consistent with the comparison between MI-RNNs and Second-order RNNs by Wu et al. (2016).

**Low-scale.** Despite the effectiveness of our models under the current hyperparameter settings, Fig. 5 reveals their limited potential when the rank is larger than 40 where the test perplexity of Vanilla-RNNs still stably decreases. Therefore, we expect that our model can outperform vanilla RNNs with a low-scale of hidden size (i.e. range from 5 to 50), but not larger scales; this is a clear tradeoff of using the simple parameterization of TT cores as we do in TTLMs.

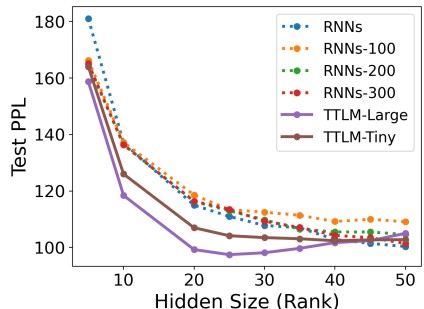

Figure 5: Comparison of test perplexity on PTB. RNNs here is Vanilla-RNNs, and its embedding size is the same as TTLM-Large and TTLM-Tiny. RNNs-100, RNNs-200 and RNNs-300 are the RNNs with fixed embedding sizes of 100, 200 and 300, respectively.

### 6.3 NON-LINEARITY ANALYSIS.

Fig. 6 shows the effects of the `tanh` activation function on TT variants based on validation PPL. Regarding the speed of convergence, `tanh` speeds up TTLM-Large-tanh, TTLM-Tiny-tanh, MI-RNNs while barely influencing second-order RNNs. Regarding the magnitude of the lowest validation perplexity, `tanh` impairs the performance TTLM-Large and TTLM-Tiny, but has little influence on multiplicative integration and the third-tensor **T** in Second-order RNNs.

Thus, the influence of non-linearity on TTLM variants depends on TT cores settings, both for the convergence of validation PPL and the magnitude of the lowest validation PPL. Thus, from an experimental point of view, the effect of non-linearity functions on one TT variant cannot simply be transferred or analogized to another TT variant. This also suggests that one should be wary of the analogy between tensor decomposition and existing neural network models at the implementation level declared by previous research (Khrulkov et al., 2018; Levine et al., 2018). The activation function could be a factor to influence such an analogy.

## 7 CONCLUSION

We first apply TT decomposition to real-world language modeling and name the framework as TTLM. We propose two variants: TTLM-Large and TTLM-Tiny, and show that they are more ef-

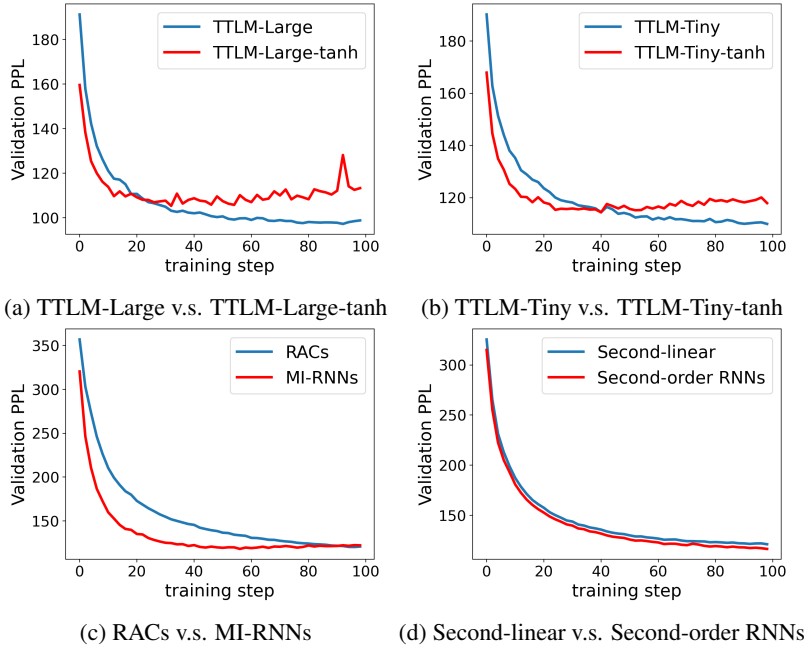

(a) TTLM-Large v.s. TTLM-Large-tanh   (b) TTLM-Tiny v.s. TTLM-Tiny-tanh

(c) RACs v.s. MI-RNNs   (d) Second-linear v.s. Second-order RNNs

Figure 6: Comparison of influence of non-linearity on TTLM variants on PTB . The suffix -tanh refers to a model using the $tanh$ activation function. Second-linear refers to Second-order RNNs without activation function.

fective compared to Vanilla-RNNs with low-scale of hidden sizes. Meanwhile, we demonstrate that Second-order RNNs, RACs and MI-RNNs are special implementations of TTLM.

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

# APPENDIX

## A  RELATIONSHIP BETWEEN TT CORES IN TTLM

To help readers understand the roles of TT cores in TTLM, we here provide a detailed calculation of the probability of a text $X = [x^{(1)}, x^{(2)}, \cdots, x^{(N)}]$ by TTLM. Note that all the intermediate TT cores are equal to each other: $\mathbf{G} = \mathbf{G}^{(2)}, ..., \mathbf{G}^{(N-1)}$ and $\mathbf{G}^{(1)} = \mathbf{G}^{(N)}$.

The calculation of $\boldsymbol{y}^{(t)}$ (i.e. the conditional probability of $x^{(t)}$ given $x^{(1:t-1)}$) at time $t$) can be described as three steps. As step I, suppose $\boldsymbol{f}(x^{(1)})$ is a one-hot vector having $f(x^{(1)})_1 = 1$. The calculation of $\mathbf{G}^{(1)}\boldsymbol{f}(x^{(1)})$ in TTLM is as follows:

$$
\begin{aligned}
\mathbf{G}^{(1)}\boldsymbol{f}(x^{(1)}) &= \begin{bmatrix} f\left(x^{(1)}\right)_1 \\ f\left(x^{(1)}\right)_2 \\ \cdots \\ f\left(x^{(1)}\right)_{|V|} \end{bmatrix} \begin{bmatrix} \mathbf{G}^{(1)}_{11} & \mathbf{G}^{(1)}_{12} & \cdots & \mathbf{G}^{(1)}_{1R} \\ \mathbf{G}^{(1)}_{21} & \mathbf{G}^{(1)}_{22} & \cdots & \mathbf{G}^{(1)}_{2R} \\ \cdots & \cdots & \cdots & \cdots \\ \mathbf{G}^{(1)}_{|V|1} & \mathbf{G}^{(1)}_{|V|2} & \cdots & \mathbf{G}^{(1)}_{|V|R} \end{bmatrix} \\
&= \left[ \mathbf{G}^{(1)}_{11}, \mathbf{G}^{(1)}_{12}, \cdots, \mathbf{G}^{(1)}_{1R} \right]^T \\
&= \boldsymbol{h}^{(1)}_{\mathrm{TTLM}}
\end{aligned}
$$

As step II, TTLM will calculate $\boldsymbol{f}(x^{(i)})\mathbf{G}\boldsymbol{h}^{(i-1)}_{\mathrm{TTLM}}$ where $i \in \{2, 3, \cdots, t-1\}$. For example, $\boldsymbol{h}^{(2)}_{\mathrm{TTLM}}$ is calculated in Eq. 7 at time $t = 2$ as follows:

$$
\boldsymbol{h}^{(2)}_{\mathrm{TTLM}} = \boldsymbol{f}(x^{(2)})^T \mathbf{G} \boldsymbol{h}^{(1)}_{\mathrm{TTLM}}
$$

As step III, TTLM will output $\boldsymbol{y}^{(t)}$ as follows:

$$
\begin{aligned}
\mathbf{G}^{(t)}\boldsymbol{h}^{(t-1)}_{\mathrm{TTLM}} &= \begin{bmatrix} \mathbf{G}^{(t)}_{11} & \mathbf{G}^{(t)}_{12} & \cdots & \mathbf{G}^{(t)}_{1R} \\ \mathbf{G}^{(t)}_{21} & \mathbf{G}^{(t)}_{22} & \cdots & \mathbf{G}^{(t)}_{2R} \\ \cdots & \cdots & \cdots & \cdots \\ \mathbf{G}^{(t)}_{|V|1} & \mathbf{G}^{(t)}_{|V|2} & \cdots & \mathbf{G}^{(t)}_{|V|R} \end{bmatrix} \begin{bmatrix} h^{(t-1)}_{\mathrm{TTLM}_1} \\ h^{(t-1)}_{\mathrm{TTLM}_2} \\ \cdots \\ h^{(t-1)}_{\mathrm{TTLM}_R} \end{bmatrix} \\
&= \begin{bmatrix} \sum_{i=1}^R \mathbf{G}^{(t)}_{1i} h^{(t-1)}_{\mathrm{TTLM}_1} \\ \sum_{i=1}^R \mathbf{G}^{(t)}_{2i} h^{(t-1)}_{\mathrm{TTLM}_2} \\ \cdots \\ \sum_{i=1}^R \mathbf{G}^{(t)}_{Ri} h^{(t-1)}_{\mathrm{TTLM}_R} \end{bmatrix}
\end{aligned}
$$

Observing the calculation, $\mathbf{G}^{(1)}$, $\mathbf{G}$ and $\mathbf{G}^{(t)}$ theoretically have no parameters in common (though we set $\mathbf{G}^{(1)} = \mathbf{G}^{(t)}$ for simplicity). Further, their roles in TTLM are different: $\mathbf{G}^{(1)}$ can be viewed as a word embedding matrix; $\mathbf{G}$ deals with two sources of information, i.e. hidden state and input word; $\mathbf{G}^{(t)}$ extracts the evidence provided in $\boldsymbol{h}^{(t-1)}_{\mathrm{TTLM}}$ and generates a set of scores over vocabulary.

## B  RELATIONSHIP BETWEEN TTLM AND SOME RNNS

We now demonstrate the relationship between TTLM and Second-order RNNs, Recurrent Arithmetic Circuits (RACs) and Multiplicative Integration RNNs (MI-RNNs).

To avoid symbol clutter when representing different RNNs, the notation is: $\boldsymbol{W}^{hx} \in \mathbb{R}^{R \times |V|}$ denotes the input-to-hidden matrix, $\boldsymbol{W}^{hh} \in \mathbb{R}^{R \times R}$ denotes hidden-to-hidden matrix, $\phi(\cdot)$ is a element-wise nonlinear activation function. Also, different hidden states are denoted as: Second-order RNNs ($\boldsymbol{h}^{(t)}_{\mathrm{2nd}}$), RACs ($\boldsymbol{h}^{(t)}_{\mathrm{RAC}}$) and MI-RNNs ($\boldsymbol{h}^{(t)}_{\mathrm{MI}}$).

## B.1 RELATION TO SECOND-ORDER RNNS

Unlike Vanilla-RNNs (Mikolov & Zweig, 2012) that have *additive* blocks, Second-order RNNs have interaction between hidden states and input data in *multiplicative* form. This is achieved by a third-order tensor $\mathbf{T}$ with the $i$-th coordinate of the hidden states $h_{2nd}^{(t)}$ defined as (Hochreiter & Schmidhuber, 1997; Maupomé & Meurs, 2020):

$$h_{2nd_i}^{(t)} = \phi(\boldsymbol{f}(x^{(t)})^T \mathbf{T}_{i,:,:} \boldsymbol{h}_{2nd}^{(t-1)} + \boldsymbol{b}) \tag{14}$$

where $\mathbf{T}_{i,:,:} \in \mathbb{R}^{M \times R}$ is the $i$th slice of tensor $\mathbf{T} \in \mathbb{R}^{M \times R \times R}$, and $\boldsymbol{b}$ is a bias vector. For simplicity, we will ignore $\boldsymbol{b}$ for other variants of RNNs since $\boldsymbol{b}$ can be seen as 0th component of $\boldsymbol{f}(x^{(t)})$ which equals to 1. Rabusseau et al. (2019) has provided that Tensor Trains can generalize linear Second-order RNNs. We here provide a basic proof from the perspective of recursive property in TTLM.

**Claim B.1.** *The third-order tensor $\mathbf{T}$ in Second-order RNNs equals the TT cores in TTLM. The hidden states of Second-order RNNs is identical to that of TTLM if they are accompanied by an activation function.*

*Proof.* The proof is based on the following observation: We recursively unfold the calculation of TTLM in Eq. 5:

$$
\begin{aligned}
\mathcal{A}_{w_1,\cdots,w_N} &= \sum_{i=1}^{|V|} f(x^{(1)})_{i_1} G_{i_1 \alpha_1}^{(1)} \cdots \\
&= \sum_{i_1,i_2=1}^{|V|} \sum_{\alpha_1=1}^{R} f(x^{(1)})_{i_1} G_{i_1\alpha_1}^{(1)} f(x^{(2)})_{i_2} G_{\alpha_1 i_2 \alpha_2} \cdots \\
&\quad\vdots \\
&= \sum_{i_1,\cdots,i_N=1}^{|V|} \sum_{\alpha_1,\cdots,\alpha_{N-1}=1}^{R} f(x^{(1)})_{i_1} G_{i_1\alpha_1}^{(1)} f(x^{(2)})_{i_2} G_{\alpha_1 i_2 \alpha_2} \cdots f(x^{(N)})_{i_N} G_{\alpha_{N-1} i_N}^{(N)}
\end{aligned}
\tag{15}
$$

Observe in the above, that at each time step, $\mathbf{G}$ has two sources of "input": the information from the previous recursive unfolding (*e.g.*, in the second line, the first line is the previous information), and the input data $\boldsymbol{f}(x^{(t)})$. From this perspective, $\mathbf{G}$ acts as a bilinear map $\mathbf{G} : \mathbb{R}^{|V|} \times \mathbb{R}^R \to \mathbb{R}^R$, and we can regard the information in the previous line as a hidden state $h_{TTLM}^{(t)}$, given by:

$$h_{TTLM_{\alpha_t}}^{(t)} = \sum_{i_t=1}^{|V|} \sum_{\alpha_t=1}^{R} f(x^{(t)})_{i_t} G_{i_t \alpha_t \alpha_{t-1}} h_{TTLM_{\alpha_{t-1}}}^{(t-1)} \tag{16}$$

where we permute the indices of $G_{\alpha_{t-1} i_t \alpha_t}$ as $G_{i_t \alpha_t \alpha_{t-1}}$ ( note that this does not change the number of indices).

We can also represent the hidden states in Second-order RNNs shown by Eq. 14 in element-wise fashion:

$$
\begin{aligned}
h_{2nd_i}^{(t)} &= \phi(\boldsymbol{f}(x^{(t)})^T \mathbf{T}_{i,:,:} \boldsymbol{h}_{2nd}^{(t-1)}) \\
&= \phi\left( \sum_{j=1}^{|V|} \sum_{k=1}^{R} f(x^{(t)})_j T_{jik} h_{2nd_k}^{(t-1)} \right)
\end{aligned}
\tag{17}
$$

where $j, k$ are the dummy indices as $i_t, \alpha_t$; $i$ specifies the coordinate of $\boldsymbol{h}_{2nd}^{(t)}$ just like $\alpha_t$ for $\boldsymbol{h}_{TTLM}^{(t)}$. Thus, $\mathbf{T}$ and $\mathbf{G}$ are the same-sized trainable bi-linear map.

After demonstrating that the third-order tensor $\mathbf{T}$ in Second-order RNNs equals the TT cores $\mathbf{G}$, the only difference between the hidden states in Eq. 17 and in Eq. 16 is an activation function. If we add an activation function for $h_{TTLM}^{(t)}$, the hidden states of Second-order RNNs and TTLM are identical, as shown in Fig. 7a.

□

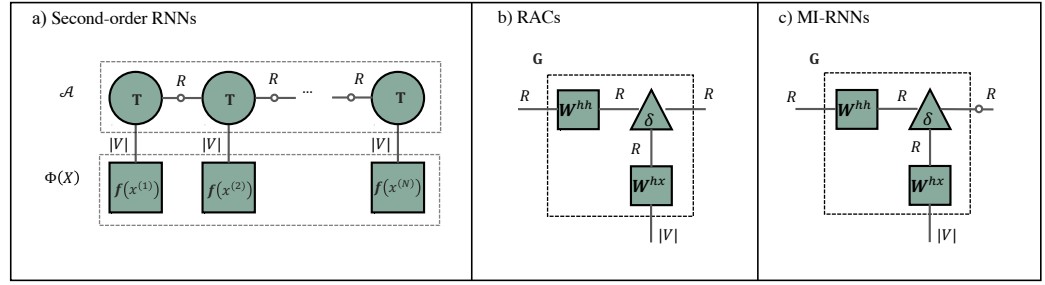

Figure 7: a) Second-order RNNs under TTLM framework. b) Hidden state of RACs under TTLM framework. c) hidden state of MI-RNNs under TTLM framework. The dashed line in the square denotes $\mathcal{A}, \Phi(X)$ or $\mathbf{G}$. The small hollow circles denote the activation functions.

## B.2 RELATION TO RACS AND MI-RNNS

We here focus on Multiplicative Integration (MI), a way to connect two sources of inputs by the Hadamard product '$\odot$'. MI has been used in RACs, Multiplicative RNNs (M-RNNs) (Sutskever et al., 2011) and MI-RNNs:

*Recurrent Arithmetic Circuits* (RACs) are recurrent networks with hidden states $h_{\text{RAC}}^{(t)}$ defined as (Levine et al., 2018):

$$h_{\text{RAC}}^{(t)} = \boldsymbol{W}^{hx}\boldsymbol{f}(x^{(t)}) \odot \boldsymbol{W}^{hh}\boldsymbol{h}_{\text{RAC}}^{(t-1)} \tag{18}$$

where these hidden states are also used as an *intermediate term* in M-RNNs.

*Multiplicative Integration RNNs* (MI-RNNs) are RACs with an activation function and hidden states $h_{\text{MI}}^{(t)}$ defined as (Wu et al., 2016):

$$h_{\text{MI}}^{(t)} = \phi(\boldsymbol{W}^{hx}\boldsymbol{f}(x^{(t)}) \odot \boldsymbol{W}^{hh}\boldsymbol{h}_{\text{MI}}^{(t-1)}) \tag{19}$$

**Claim B.2.** *Given the condition the TT-scores:* $\mathbf{G} = \boldsymbol{W}^{hx} \odot \boldsymbol{W}^{hh}$. *The hidden states of RACs are identical to that of TTLM. The hidden states of MI-RNNs are identical to that of TTLM if they are accompanied by an activation function.*

*Proof.* The proof is based on the following observation: In the language of tensor contractions, Eq. 18 involves contracting the input weights matrix $\boldsymbol{W}^{hx}$ with the input vector $\boldsymbol{f}(x^{(t)})$, and contracting the hidden weights matrix $\boldsymbol{W}^{hh}$ with $h_{\text{RAC}}^{(t-1)}$. The Hadamard product of the two is a third-order diagonal tensor $\delta \in \mathbb{R}^{R \times R \times R}$ such that $\delta_{ijk} = 1$ iff the $i = j = k$, and $\delta_{ijk} = 0$ otherwise. Thus, we can write Eq. 18 in element-wise fashion:

$$
\begin{aligned}
h_{\text{RAC}_{\alpha_t}}^{(t)} &= \sum_{i_t=1}^{|V|} \sum_{\alpha_t=1}^{R} f(x^{(t)})_{i_t} W_{i_t j}^{hx} \delta_{j\alpha_t k} W_{k\alpha_{t-1}}^{hh} h_{\text{RAC}_{\alpha_{t-1}}}^{(t-1)} \\
&= \sum_{i_t=1}^{|V|} \sum_{\alpha_t=1}^{R} f(x^{(t)})_{i_t} \mathbf{G}_{i_t \alpha_t \alpha_{t-1}} h_{\text{RAC}_{\alpha_{t-1}}}^{(t-1)}
\end{aligned}
\tag{20}
$$

where $\mathbf{G} = \boldsymbol{W}^{hx} \odot \boldsymbol{W}^{hh}$. In this case, the hidden state of TTLM in Eq. 16 is equal to the hidden state of RACs in Eq. 20, as shown in Fig. 7b. Similarly, if Eq. 16 is accompanied with an activation function $\phi$, Eq. 16 is equal to the hidden state of MI-RNNs in Eq. 19 as shown in Fig. 7c.

$\square$

Given Claim B.1 and B.2, the three models shall be simulated by TTLM with a non-linear activation function and we leave finding a theoretical proof of this conjecture to a future work.

## C  IMPLEMENTATIONS

We implement all RNNs models, TTLM, TTLM-Large, and TTLM-Tiny using PyTorch. The weights in the models are adjusted to minimize the average cross entropy loss over training sequences via stochastic gradient descent computed using the truncated backpropagation through time algorithm. (Werbos, 1990; Williams & Peng, 1990)).

For RNNs, there are five matrix parameters: $\boldsymbol{W}^{xe} \in \mathbb{R}^{E \times |V|}$ is the input embedding matrix, $\boldsymbol{W}^{eh} \in \mathbb{R}^{E \times H}$ is the embedding-to- hidden matrix, $\boldsymbol{W}^{hh} \in \mathbb{R}^{H \times H}$ is the hidden-to-hidden matrix. We tie (share the same training parameters) the input embedding $\boldsymbol{W}^{xe}$ and output embedding $\boldsymbol{V}$ which has been proved lead to a significant reduction in perplexity (Press & Wolf, 2016). So there is a projection matrix $\boldsymbol{P} \in \mathbb{R}^{H \times E}$ before the output embedding. All this process is introduced in (Press & Wolf, 2016).

For TTLM models, we tie the input tensor $\mathbf{W}^{xe}$ and $\mathbf{V}$. The implementation of $\delta$ is functioned by a reshape function, so the interaction between hidden and input can be computed by matrix product. We also let $\mathbf{G}^{(1)}$ have the same parameters along the dimension $|V|$ (i.e. $\mathbf{G}^{(1)}$ is simplified into a $\boldsymbol{G}^{(1)} \in \mathbb{R}^{1 \times R}$ and can be viewed as initial hidden state $\boldsymbol{h}^{(0)}$).

| Model | Training Parameters |
|---|---|
| Vanilla-RNN | $\boldsymbol{W}^{xe} \in \mathbb{R}^{E \times |V|}, \boldsymbol{W}^{eh} \in \mathbb{R}^{E \times H}, \boldsymbol{W}^{hh} \in \mathbb{R}^{H \times H},$ $\boldsymbol{P} \in \mathbb{R}^{H \times E}, \boldsymbol{V} \in \mathbb{R}^{E \times |V|}$ |
| MI-RNNs | $\boldsymbol{W}^{xe} \in \mathbb{R}^{E \times |V|}, \boldsymbol{W}^{eh} \in \mathbb{R}^{E \times H}, \boldsymbol{W}^{hh} \in \mathbb{R}^{H \times H},$ $\boldsymbol{P} \in \mathbb{R}^{H \times E}, \boldsymbol{V} \in \mathbb{R}^{E \times |V|}$ |
| RACs | $\boldsymbol{W}^{xe} \in \mathbb{R}^{E \times |V|}, \boldsymbol{W}^{eh} \in \mathbb{R}^{E \times H}, \boldsymbol{W}^{hh} \in \mathbb{R}^{H \times H},$ $\boldsymbol{P} \in \mathbb{R}^{H \times E}, \boldsymbol{V} \in \mathbb{R}^{E \times |V|}$ |
| Second-order RNNs | $\boldsymbol{W}^{xe} \in \mathbb{R}^{E \times |V|}, \mathbf{T} \in \mathbb{R}^{H \times H \times H}, \boldsymbol{W}^{hh} \in \mathbb{R}^{E \times H},$ $\boldsymbol{P} \in \mathbb{R}^{H \times E}, \boldsymbol{V} \in \mathbb{R}^{E \times |V|}$ |
| TTLM | $\mathbf{G} \in \mathbb{R}^{R \times |V| \times R}, \boldsymbol{G}^{(t)} \in \mathbb{R}^{R \times |V|}, \boldsymbol{G}^{(1)} \in \mathbb{R}^{R \times |V|}$ |
| TTLM-Tiny | $\mathbf{W}^{xe} \in \mathbb{R}^{R \times |V| \times R}, \boldsymbol{W}^{hh} \in \mathbb{R}^{R \times R},$ $\mathbf{P} \in \mathbb{R}^{R \times R \times R}, \mathbf{V} \in \mathbb{R}^{R \times R \times |V|}$ |
| TTLM-Large | $\mathbf{W}^{xe} \in \mathbb{R}^{R \times |V| \times R}, \mathbf{W}^{eh} \in \mathbb{R}^{R \times R \times R \times R}, \boldsymbol{W}^{hh} \in \mathbb{R}^{R \times R},$ $\mathbf{P} \in \mathbb{R}^{R \times R \times R}, \mathbf{V} \in \mathbb{R}^{R \times R \times |V|}$ |

Table 2: Training parameters in our implementation. $E$ is the embedding size, $H$ is the hidden size in RNNs, and $R$ is the rank in the TTLM. We set $H = R$ and $E = R^2$ to make the parameters of all models in the same scale. The parameters of $\boldsymbol{W}^{xe}$ and $\mathbf{W}^{xe}$ are uniformly initialized in the interval $[-0.1, 0.1]$, $\boldsymbol{W}^{eh}$, $\mathbf{W}^{eh}$ and $\boldsymbol{W}^{hh}$ are uniformly initialized between $[-\frac{1}{\sqrt{H}}, \frac{1}{\sqrt{H}}]$.

