# OpenReview forum: "Language Modeling Using Tensor Trains"
_ICLR.cc/2023/Conference — Submitted to ICLR 2023_

### Official Review · Reviewer_K2sM · 2022-10-23

**Confidence:** 4
**Correctness:** 1
**Technical Novelty And Significance:** 1
**Empirical Novelty And Significance:** 2
**Recommendation:** 1

**Clarity, Quality, Novelty And Reproducibility:**

**--- Clarity/Quality ---**

The clarity and quality of writing is poor. This was discussed under weaknesses.

**--- Novelty ---**

It's not clear how the proposal is different from previous works. The claimed theoretical contributions are incorrect.

**--- Reproducibility ---**

Since it's very hard to follow the text, and no code is provided, the results aren't reproducible.

**Strength And Weaknesses:**

**--- Strengths ---**

No particular strengths stood out to me when reviewing this paper.

**--- Weaknesses ---**

**W1.** The paper claims (e.g., in the abstract and the conclusion) to show that the proposed TTLM is a generalization of 2nd-order RNNs, RACs, and MI-RNNs. The TTLM model takes the form given in Eq (2). In order to say that TTLM generalizes another model, that model must be expressible in the form given in Eq (2). However, according to Claim 4.1 and 4.2, in order to be able to express 2nd-order RNNs and MI-RNNs additional nonlinearities need to be inserted into the formula in Eq (2), therefore yielding a model which is *not* expressible as a TTLM (i.e., as in Eq (2)). Therefore, it is *incorrect* to say that TTLM generalizes those models! If TTLM truly generalized these models, there shouldn't be a need to further modify Eq (2).

This is a bit like saying that the set of linear functions (i.e., matrices) generalize functions $f : \mathbb{R}^{m} \rightarrow \mathbb{R}^{n}$ of the form $f(x) = \sigma(A x)$, where $A$ is a matrix $\sigma$ is a non-linear functions.

**W2.** The paper is full of typos, missing/redundant commas and periods, and notational inconsistencies that a careful proof reading should have caught; there are two inconsistencies already in the first sentence in Sec 1.

**W3.** On page 1, you say that "we propose a novel Tensor Train Language Model, as a first attempt to apply tensor networks on language modeling tasks". This makes it sound like applying tensor networks, and tensor trains in particular, to language modeling is new. But that's not the case. For example, Miller et al. (2021) use matrix product states (which is the same as tensor trains) with language modeling as a potential application. Your contribution needs to be clarified.

**W4.** The paper is difficult to follow in several places. For example:
- The discussion in Sec 3.3 doesn't make sense. You insert Eq (4) into Eq (3) to get Eq (5)---which is the same as Eq (3). What is the point of this calculation?
- Below Eq (5) you show diagrammatic notation and say that it represents the tensor $A$---but doesn't that tensor network graph correspond to the contraction in Eq (2)?
- The discussion at the start of Sec 4 is difficult to follow. TNLM is never defined.
- How exactly is the slice operator $T[i]$ defined? This is unclear below Eq (6). Is the 2nd or 3rd index kept fixed when slicing?
- In Fig 3: Plot (a) shows that $f_\theta(x^{(j)})$ is contracted with the tensor $M^{(j)}$. But then (b) and (c) seem to indicate that $M^{(j)}$ itself is a reshaped version (a matrix, rather than tensor) of a contraction that involves $f_\theta(x^{(j)})$. Also, since TTML-Large and TTML-Tiny add additional operations like reshape, it's not immediately clear that they can even be expressed on the form in Eq (2). Again, if additional functions need to be added into Eq (2) to make the model expressible in that format, then it's not a TTLM.
- In Fig 4: The caption refers to labels (RNNs-100, RNNs-200, etc) that don't appear in the figure.

**W5.** The experiment results aren't represented properly. For example, you say that "TTLM-Large obtains the best PPL among these models," but that's not true. Two of the LSTM-based methods achieve lower PPL in Table 1 (for PTB). For the PTB dataset, the PPL for TTLM-Large is bolded even though it's not the best number.

**Summary Of The Paper:**

The paper proposes a model for language modeling. The proposed method models the probability of a certain word sequence as the inner product between a feature vector which is a tensor product of feature vectors for each word, and a learnable weight tensor which is in tensor train (TT) format. The paper claims to show that the proposed method is a generalization of three other RNN-based language models (it's not in two of the three cases), and also runs some numerical experiments.

**Summary Of The Review:**

The proposed method is poorly explained. The claimed theoretical results are incorrect. The paper is full of typos and notational inconsistencies. This is a clear reject for me.

---

> ### Author Response · Authors · 2022-11-17
> **To review K2sM**
>
> Thank you all the same. We have answered most of your questions in the new version, and we would be grateful if you could look at our new version, as the first version we wrote in haste may have clouded your judgment.
> Below we detail each comment individually.
>
>
> > The paper claims (e.g., in the abstract and the conclusion) to show that the proposed TTLM is a generalization of 2nd-order RNNs, RACs, and MI-RNNs. The TTLM model takes the form given in Eq (2). In order to say that TTLM generalizes another model, that model must be expressible in the form given in Eq (2). However, according to Claim 4.1 and 4.2, in order to be able to express 2nd-order RNNs and MI-RNNs additional nonlinearities need to be inserted into the formula in Eq (2), therefore yielding a model which is not expressible as a TTLM (i.e., as in Eq (2)). Therefore, it is incorrect to say that TTLM generalizes those models! If TTLM truly generalized these models, there shouldn't be a need to further modify Eq (2). This is a bit like saying that the set of linear functions (i.e., matrices) generalize functions  of the form , where  is a matrix  is a non-linear functions.
>
>
> We have toned it in our new version. In our new paper version, we just stated: we demonstrate the relationship between TTLM and Second-order Recurrent Neural Networks (RNNs), Recurrent Arithmetic Circuits, and Multiplicative Integration RNNs.
>
>
> > On page 1, you say that "we propose a novel Tensor Train Language Model, as a first attempt to apply tensor networks on language modeling tasks". This makes it sound like applying tensor networks, and tensor trains in particular, to language modeling is new. But that's not the case. For example, Miller et al. (2021) use matrix product states (which is the same as tensor trains) with language modeling as a potential application. Your contribution needs to be clarified.
>
> We have clarified our contribution in our new version (Sec. 2). Note that they fail to use matrix product states for language modeling tasks due to some of the obstacles posed by their model.
>
> > The discussion in Sec 3.3 doesn't make sense. You insert Eq (4) into Eq (3) to get Eq (5)---which is the same as Eq (3). What is the point of this calculation?
>
> Sorry for not being clearer. We have revised it in the new version. This part is to show that Eq (5) can be a language modeling by the inner product of tensor $\mathcal{A}$ and $\Phi(x)$ at low-dimensional space. Eq (3) is just a Tensor Train decomposition of the high-dimensional tensor, but it is not a language model (it does not contain "input data").
>
> > Below Eq (5) you show diagrammatic notation and say that it represents the tensor ---but doesn't that tensor network graph correspond to the contraction in Eq (2)?
>
> Eq (5) is the TT format of Eq (2). Thus, their diagrammatic notations should be the same.
>
>
> > How exactly is the slice operator defined? This is unclear below Eq (6). Is the 2nd or 3rd index kept fixed when slicing?
>
> This is a common mathematical usage in this area [On Multiplicative Integration with Recurrent Neural Networks](https://arxiv.org/pdf/1606.06630.pdf).
>
>
> > The experiment results aren't represented properly. For example, you say that "TTLM-Large obtains the best PPL among these models," but that's not true. Two of the LSTM-based methods achieve lower PPL in Table 1 (for PTB). For the PTB dataset, the PPL for TTLM-Large is bolded even though it's not the best number. TTLM-Large obtains the best PPL among these models
>
> Sorry, we mean in the paper that TTLM-Large obtains the best PPL among our implemented models (baselines)  under the same training environments.

---

### Official Review · Reviewer_meG2 · 2022-10-23

**Confidence:** 4
**Correctness:** 3
**Technical Novelty And Significance:** 2
**Empirical Novelty And Significance:** 2
**Recommendation:** 5

**Clarity, Quality, Novelty And Reproducibility:**

Clarity: the paper is well presented with rich algebra background.

Quality: the quality if this work is good

Novelty: the novelty of this work is not very high since Tensor Train has been investigated for many year.

Reproducibility: unclear, it is not easy to implement the algorithm based on the paper only.

**Strength And Weaknesses:**

Strength
1. the paper is well written, containing all of the Algebra background of tensor network and tensor train.

2.  they prove that TTLM generalizes Second-order RNNs, RACs and MI-RNNs.

Weakness,
1. Applying Tensor Train to RNN is not very new. For example,

Yang, Yinchong, Denis Krompass, and Volker Tresp. "Tensor-train recurrent neural networks for video classification." International Conference on Machine Learning. PMLR, 2017.

Actually, TT is a very common compression technique, and is widely used with CNN, Transformer.
Ma, Xindian, et al. "A tensorized transformer for language modeling." Advances in neural information processing systems 32 (2019).

Therefore, the novelty of this work is not very high.

2. The code is not available, and the reproducibility is uclear due to a lots of details in the proposed framework.

3. It may be better to compare with the popular Transformer to see more benefit of the TTML

**Summary Of The Paper:**

In this work, the authors propose a Tensor Train Language Model (TTLM) to apply tensor networks for language modelling. Also, they prove that TTLM generalizes Second-order RNNs, RACs and MI-RNNs.

**Summary Of The Review:**

In this work, the authors propose a Tensor Train Language Model (TTLM) to apply tensor networks for language modelling. Also, they prove that TTLM generalizes Second-order RNNs, RACs and MI-RNNs.

However, the novelty of this work is not very high and the reproducibility is uclear.

---

> ### Author Response · Authors · 2022-11-17
> **To reviewer meG2**
>
> Thanks for your comments on our paper. Here, we will answer your questions in detail.
> > Applying Tensor Train to RNN is not very new. For example
> "Tensor-train recurrent neural networks for video classification"
> "Actually, TT is a very common compression technique, and is widely used with CNN, Transformer. Ma, Xindian, et al. "A tensorized transformer for language modeling." Advances in neural information processing systems 32 (2019)."
>
> We admit that tensor train technologies have been applied to machine learning or video classification. However, our work is the first step to applying tensor networks to **real-world language modeling** tasks. The importance of our work is what reviewer CPvM emphasized, "Giving concrete experimental results in language modeling is an important step for research into the use of the tensor network. the prior work in this area has skewed heavily towards theory".
>
> > The code is not available, and the reproducibility is uclear due to a lots of details in the proposed framework.
> Thanks for your advice.
>
> The code has been available at [https://github.com/tensortrainlm/tensortrainlm]( https://github.com/tensortrainlm/tensortrainlm).
>
> > It may be better to compare with the popular Transformer to see more benefits of the TTLM.
>
> We hope it can be future work. As proof-of-concept work, we aim to apply a tensor network to a real-world language modeling task.

---

### Official Review · Reviewer_CPvM · 2022-10-26

**Confidence:** 4
**Correctness:** 3
**Technical Novelty And Significance:** 2
**Empirical Novelty And Significance:** 3
**Recommendation:** 5

**Clarity, Quality, Novelty And Reproducibility:**

* Not including the size of the embedding layer in the model size comparison in Table 1 feels misleading, as the TTLM models will have these embedding layers scale much more rapidly with the number of hidden units R than the other models on the table. In particular, the embedding layer for the TTLMs will have $|V| R^2$ parameters, whereas other models will require $|V| R$ or less. Given that the vocabulary is much larger than any of the model parameters here, this represents a significant memory and runtime overhead in larger models that isn't remarked on anywhere in the paper. I would strongly encourage the authors to include this in their parameter count in Table 1, either as a part of the model size or as a separate column.

* In light of the point above, the many references to the TTLM-Tiny model using "half the parameters of vanilla RNNs" should probably be toned down. Would this comparison remain as favorable if the value of R is increased? If not, I would encourage the authors to not use this as a selling point of their model.

* The paper has many points that could be made more clear, and I list some of these below in the form of questions or unexplained points that I encountered while reading the paper.

    * How does the TTLM ensure its probabilities are actually non-negative numbers? The function g(X) in Eq. (2), which together with Eq. (5) determines the probabilities assigned to text X, can be positive or negative depending on the value of the TT cores, and no strategy is mentioned for handling the possibility of negative values.
    * How does the TTLM ensure its probabilities are properly normalized? This point is also not remarked on in the paper.
    * What is the hidden unit count (i.e. the value of R) used to get the results in Table 1?
    * What vocabulary size was used for each experiment? Is the vocabulary formed from individual words, individual characters, or something else (e.g. tokens resulting from a different tokenization process)?
    * How are the parameters of the TTLM model initialized?

* It is confusing that the TTLM is introduced in Section 3.3 and Figure 2 as using site-dependent TT cores $G^{(i)}$, when in fact all cores are chosen the same in later sections of the paper.

* As a small suggestion, it would seem more natural to describe the operation of the TTLMs in Figure 3 as mapping the one-hot encoded vector $f_\theta(x^{(t)})$ directly to an RxR matrix $M^{(t)}$, rather than to an $R^2$ dimensional vector which is reshaped into a matrix. This is trivial for the TTLM-Tiny model (this already appears in Figure 2a), and can be done for the TTLM-Large model by depicting $W^{eh}$ as a fourth-order tensor mapping RxR matrices to other RxR matrices.

* The paper has multiple small typos, and I would encourage the authors to use a spelling and grammar checker to fix some of the more obvious issues along these lines.

**Strength And Weaknesses:**

+ Giving concrete experimental results in language modeling is an important step for research into the use of TNs in machine learning. These models have many interesting properties and capabilities that aren't shared by neural nets, but (largely owing to the origins of TNs in physics and mathematics) prior work in this area has skewed heavily towards theory. I want to stress to other reviewers that even though the empirical results reported are a far cry from that of modern neural language models, the fact that simple recurrent TT models are in the same ballpark as pre-Transformer state of the art RNNs is an important result in itself.

+ The authors prove concrete connections between recurrent TT language models and several prior models, namely recurrent arithmetic circuits, second-order RNNs, and multiplicative integration RNNs. These connections aren't surprising and the proofs are basic, but it is nonetheless good to clearly state these connections so that they can be appreciated by the broader machine learning community.

- Many important implementational details are unspecified or unclear, and the paper feels hastily written. I address these points in more detail in the following section, but given that the primary contribution of this work is experimental, ensuring that these experiments are well-explained and fully reproducible should be a priority.

**Summary Of The Paper:**

The paper gives an experimental characterization and some theoretical connections regarding language models built on recurrent tensor train (TT) models, a form of tensor network (TN). Although TNs have received an increasing amount of interest within machine learning, with TTs having been proposed as promising language models, this appears to be the first work to actually evaluate the performance of such models in language modeling.

**Summary Of The Review:**

The experimental contributions of the paper are important, and the theoretical results prove several useful relationships between previous model families. However, the lack of clarity in the description of the model and experiments, along with the absence of significantly novel architectural or theoretical contributions, detract from the paper's score.

---

> ### Author Response · Authors · 2022-11-17
> **To reviewer CPvM**
>
> Thanks for your feedback on our paper, and we really like your comments.  We have answered most of your questions in the new version, and below, we detail each comment individually.
>
> >Not including the size of the embedding layer.
>
> Now, we have included the experimental details in Section 6. the embedding size is 400, and the hidden size and rank of TTLM are 20.
>
> >In light of the point above, the many references to the TTLM-Tiny model using "half the parameters of vanilla RNNs" should probably be toned down. Would this comparison remain favorable if the value of R is increased? If not, I would encourage the authors not to use this as a selling point of their model.
> >
> Good suggestion. We have rewritten this part, and we don't emphasize "our model size is half the parameters of vanilla RNNs" in our new version.
> > How does the TTLM ensure its probabilities are non-negative numbers?
> >
> We have stated how we can compute the probabilities in Section 4.4 in our new version.
>
> > It is confusing that the TTLM is introduced in Section 3.3 and Figure 2 as using site-dependent TT cores when in fact, all cores are chosen the same in later sections of the paper.
>
> We have revised it in our new version. In section 4.3, we now have a clear definition of TTLM.
>
> > As a small suggestion, it would seem more natural to describe the operation of the TTLMs in Figure 3 as mapping the one-hot encoded vector directly to an RxR matrix
> Good suggestion.
>
> We have rewritten this part. In section 5.1, we introduce a fourth-order tensor $\delta$.
>
> > The paper has multiple small typos, and I would encourage the authors to use a spelling and grammar checker to fix some of the more obvious issues along these lines.
>
> We admit we wrote our paper hastily. We appreciate your taking the time to provide these constructive suggestions. We have conducted the spelling check issues in this new version.

---

### Public Comment · ~Guillaume_Rabusseau1 · 2022-11-06
**Missing references**

The authors seem to have missed important references where the fact that TTLM generalize linear second order RNN has been discussed and proved: [1] and [2] (see Figure 2 and Eq 2 in [1] and Section 3 in [2] where the equivalence between WFA ---which are equivalent to second order RNN with linear activation functions--- and TT is extensively discussed).

[1] Rabusseau, Guillaume, Tianyu Li, and Doina Precup. "Connecting weighted automata and recurrent neural networks through spectral learning." The 22nd International Conference on Artificial Intelligence and Statistics. PMLR, 2019.

[2] Li, Tianyu, Doina Precup, and Guillaume Rabusseau. "Connecting weighted automata, tensor networks and recurrent neural networks through spectral learning." Machine Learning (2022): 1-35.

---

> ### Author Response · Authors · 2022-11-17
> **To Guillaume Rabusseau**
>
> Thank you for pointing out these two papers. We have cited your work in the current revision and clarified our contribution.

---

### Author Response · Authors · 2022-11-17
**We sincerely thank you for providing comments and reviews, and look forward to more feedback.**

Dear Reviewers,

We thank all the reviewers for their insightful and valuable comments (Especially thank the reviewer $CPvM$).

Based on the first-round reviews, we have revised our paper to respond to your concerns and comments.

1. We have reorganized the illustration about the TTLM in Sec. 4. We clarify the definition of TTLM, including how it theoretically computes the probability of a given text.

2. In the revised version, we provide our code (Abstract) and implementation details, including model parameters (Appendix C), tasks, and hyper-parameters (Sec. 6.1). We also removed the emphasis on model parameters, and have further explained the difference between parameters of models ( Appendix C).

3. We provide a new section (Sec. 2) to clarify our theoretical contribution (we first derive a tensor network language model in the way that it can be applied to real-world language modeling datasets). We also revise many sentences to avoid ambiguous words (such as "generalize").

4. We have provided a new version of introducing our two proposed models, TTLM-Large and TTLM-Small (thanks to $CPvM$) (Sec. 5.1).

6. We have fixed many typos and unclear text. We apologize for any inconvenience this may have caused previously)

---

### Author Response · Authors · 2022-12-05
**We are looking forward to your feedback on the new version**

Since the discussion deadline diary is coming up (Dec 12  2022), we would like to express once again that we are very much looking forward to receiving new comments on the revised version.

---

### Decision · Program_Chairs · 2023-01-20

**Decision:**

Reject

**Justification For Why Not Higher Score:**

The paper is not yet ready: it does not have the killer feature for the proposed methods and could benefit from a better tensor baseline, as well as positioning among applications and methods.

**Justification For Why Not Lower Score:**

N/A

**Metareview: Summary, Strengths And Weaknesses:**

The paper proposes to model conditional probabilities of the sequence using a tensor-train based model with additional softmax applied to the "hidden state" in order to achieve non-negativity and sum equal to 1 for the conditional probabilities.  The resulting model is tested on several benchmark examples:

Strengths: The paper is well-written and easy to follow, and the method seems to work in a proper way as expected.

Weaknesses: TT-model has several properties that make it quite beneficial (i.e., computation of the marginals), which are not clear with the softmax model for conditional probabilities. A similar model for sequence modelling directly in the TT-format has been proposed in "Tensor-train density estimation" (Novikov, Panov, Oseledets, AISTATS 2021), which could be used as a baseline.
Another weakness is that in the applications selected there are much stronger baselines (Transformers), so the TTLM should beat them at least in something (training time, size, etc.). Otherwise, the benefit is not clear.

Overall, the reviewers were not very happy with the current version, but I encourage the authors to take the constructive comments into the account and submit to the next venue, since the paper does have the potential.



**Summary Of Ac-Reviewer Meeting:**

N/A